# Edible unclonable functions

Jung Woo Leem [1], Min Seok Kim [2], Seung Ho Choi[3], Seong-Ryul Kim[4], Seong-Wan Kim[4], Young Min Song[2], Robert J. Young[5] & Young L. Kim[1,6,7,8]*

Counterfeit medicines are a fundamental security problem. Counterfeiting medication poses a tremendous threat to patient safety, public health, and the economy in developed and less developed countries. Current solutions are often vulnerable due to the limited security levels. We propose that the highest protection against counterfeit medicines would be a combination of a physically unclonable function (PUF) with on-dose authentication. A PUF can provide a digital fingerprint with multiple pairs of input challenges and output responses. On-dose authentication can verify every individual pill without removing the identification tag. Here, we report on-dose PUFs that can be directly attached onto the surface of medicines, be swallowed, and digested. Fluorescent proteins and silk proteins serve as edible photonic biomaterials and the photoluminescent properties provide parametric support of challenge-response pairs. Such edible cryptographic primitives can play an important role in pharmaceutical anti-counterfeiting and other security applications requiring immediate destruction or vanishing features.

[1] Weldon School of Biomedical Engineering, Purdue University, West Lafayette, Indiana 47907, USA. [2] School of Electrical Engineering and Computer Science, Gwangju Institute of Science Technology, Gwangju 61005, Republic of Korea. [3] Department of Biomedical Engineering, Yonsei University, Wonju 26493, Republic of Korea. [4] Department of Agricultural Biology, National Institute of Agricultural Sciences, Rural Development Administration, Wanju 55365, Republic of Korea. [5] Department of Physics, Lancaster University, Lancaster LA1 4YB, UK. [6] Purdue University Center for Cancer Research, West Lafayette, Indiana 47907, USA. [7] Regenstrief Center for Healthcare Engineering, West Lafayette, Indiana 47907, USA. [8] Purdue Quantum Science and Engineering Institute, West Lafayette, Indiana 47907, USA. *email: youngkim@purdue.edu

The problem of counterfeit medicines is not a new one, but is becoming a tremendous burden to society in all countries. While fake pharmaceutical products can be explicitly categorized into substandard, falsified, counterfeit, and diverted ones[1], they are all often referred to, as a single group, as counterfeit medicines. Counterfeit medicines pose a significant threat to patient safety and public health, as well as cause heavy economic losses in developed and less developed countries. As a devastating example, counterfeit drugs for malaria and pneumonia treatments cause estimated 250,000 child deaths each year[2]. Counterfeit medicines of both lifestyle drugs (e.g., treatments for erectile dysfunction) and lifesaving drugs (e.g., treatments for cancer, malaria, diabetes, etc.) are increasingly being produced in developed and developing countries, in part due to the increased public use of online pharmacies[3–8]. In addition, as an infringement of intellectual properties, scientific innovations and financial rewards in pharmaceutical companies are undermined by the widespread counterfeiting of medicines[5]. The health and economic consequences of counterfeit medicines are far more serious in low-income and middle-income countries. It is estimated that counterfeit medicines account for 10% of the global pharmaceutical trade and more than 20–30% of all medicines in Africa, Asia, and the Middle East[6,9].

There are a variety of approaches for detecting counterfeit medicines and for offering possible solutions for reducing the threat[1,10–12]. Traditionally, analytical chemistry and spectroscopy technologies have been used to identify counterfeit medicines by detecting chemical signatures of major ingredients. Marking and printing on the medicine surface are incorporated at various levels of resolution using lasers and other proprietary technologies, which modify the outer surface or coating of tablets or capsules. Recently, digital anti-counterfeit technologies have played a more significant role in authentication and supply chain[1,10–12]. Package-level barcodes and radio frequency identification (RFID) are commonly used for instantaneous remote authentication. Several mobile technologies have been introduced for authentication services, track and trace solutions, and medicine recognitions. Detrimentally, such authentication and security techniques are symmetric; if illegitimate manufacturers or sellers have access to the same techniques, it would be possible for them to create clones. An ideal authentication technology should be asymmetric with a form of on-dose authentication which can be directly swallowed and digested. Specifically, on-dose (or in-dose) authentication means that every individual pill or dose is verified as genuine in the absence of packaging and one can ingest it. Even if the original packaging is not retained by pharmacists or patients, the possibility of ingestion of counterfeit medicines is maximally minimized. Indeed, the packaging information is often unavailable; pills are sold in small quantities and individual strips dispensed by pharmacists. On-dose authentication maximally reduces the opportunity for illegitimate sellers to use expired, counterfeit, or substandard drugs.

In this respect, a few promising technologies have recently been introduced with the potential of digital authentication, including digitally encoded polymers[13], QR-coded microtaggants and advanced wrinkle-based tags[14,15], QR code printing of active pharmaceutical ingredients[16], encoded-multifunctional hydrogel microparticles[17], large-scale microparticle arrays[18], encoded metal nanomaterials[19,20], and silica microtags[21]. However, such materials are often not ideal from an oral intake safety perspective. These approaches rely on biocompatible and biodegradable yet exogenous materials, such as polystyrene, cellulose-acetate-phthalate (CAP), poly(lactic-co-glycolic acid) (PLGA), poly(ethylene glycol) (PEG), poly(ethylene glycol) diacrylate (PEGDA), silver, gold, and silica. On the other hand, it should be noted that foreign and nano-sized food additives could potentially have hazardous and adverse (e.g., carcinogenic and cytotoxic) effects, which currently result in limited utilizations[22–24]. In addition, such simple tagging technologies could be vulnerable to attackers, due to the limited security level[25].

One excellent way for guaranteeing the high security of on-dose authentication and protection against counterfeiting medicines is to take advantage of physically unclonable functions (PUFs) that were originally developed for hardware and information security. PUFs provide digital fingerprints where information is usually read from a static entropy (physical disorder) embedded in the system. Thus, PUFs play a critical role in identifying and authenticating devices, objects, programs, and data[26–29]. PUFs should be distinguished from unique objects or security tags such that PUFs have interactive pairs of input challenges and output responses. Upon being reacted by an input external stimulus (i.e., challenge), the PUF generates a corresponding output response. Depending on PUF types and applications, the number of challenge-response pairs ranges from a few to an exponential number of pairs. Importantly, PUFs can be asymmetric such that it is easy to make a PUF, but is extremely challenging for counterfeiters to create a clone. Once an output response is read from the database, it cannot be re-used as well. The information on dose, frequency, and caution can be encoded for user adherence by labeling individual medicines. Although the majority of focus in the PUF community revolves around various architectures of electronic PUFs[26–29], other advanced PUFs have been proposed and introduced, including nanoelectronics, photonics, and chemical approaches[30–40]. On the other hand, PUFs have not yet been translated for pharmaceutical applications to the best of our knowledge.

For digital on-dose PUFs, we propose to use silk proteins and fluorescent proteins as edible and digestible photonic biomaterials. From an edible perspective, the key requirements include digestibility and nonallergenic potential. Although there are recent advances in biocompatible materials, these still rely on exogenous materials that often result in severe immune and inflammatory responses in the body[41]. Endogenous natural materials or biomaterials would be ideal for on-dose applications. Importantly, silk proteins (i.e., fibroin) have excellent intrinsic functionality, biocompatibility, and low immunogenicity with minimal inflammatory and immune responses[42–46]. Naturally derived silk fibroin, without any external treatment, can be dissolved in an aqueous solution[47–49]. Silk proteins are also degradable and the degradation rate is controllable by using different silk regeneration and fabrication methods. More relevantly, silk proteins are edible and digestible[49–51]. In addition, fluorescent proteins have been introduced into the food supply from genetically modified food[52]. The potential toxicity and allergenicity are minimal with ingestion of green fluorescent protein. When compared with common food allergens, fluorescent proteins do not have common allergen epitopes and are degraded during gastric digestion[53,54]. For an engineering perspective, processing of silk proteins is readily available for constructing structures and patterns from nanoscale to microscale[42,43,55,56]. In particular, the polymeric nature from silk proteins can easily be fabricated into a variety of types of rigid or flexible structures with tunable mechanical and optical properties[42,43,48]. Furthermore, silk proteins containing recombinant fluorescent proteins can be produced by transgenesis of genetically engineered domesticated silkworms. Transgenes of a variety of fluorescent proteins can be expressed by germline transformation using the gene splicing method *piggyBac*[45,46,55,57–59]. The hybridization method can yield transformed silkworms with multiple successive generations and produce fluorescent silk in large amounts.

We take use of particulate fluorescent silk microparticles. In this case, the source of entropy can be a randomly scattered (or broadcast) fluorescent microparticle admixture, as the behavior of granular or particulate materials intrinsically has complex

spatiotemporal fluctuations during the fabrication process[60,61]. For the challenge-response requirement of PUFs, a unique set of excitation and emission bands of different fluorescent proteins can serve as input challenges. Then, images of spontaneous emission (i.e., fluorescence) can generate genuinely inherent output responses. For practical and reliable PUFs, it is important to guarantee fundamental PUF performance, including bit uniformity, device uniqueness (for security), and readout reproducibility (for reliability)[62]. In brief, the bit uniformity of a PUF measures the probability of observing 1-bit or 0-bit states in the response bits. The ideal bit uniformity is 50%. The device uniqueness means that each PUF is uniquely distinguishable among a group of PUFs manufactured under the same conditions. The readout reproducibility measures how reliable a PUF is in reproducing the same bits when the PUF is read multiple times. When such fundamental performance requirements are met, a PUF can potentially provide an immediate solution for authentication and anti-counterfeiting with high security.

In this paper, we introduce the combination of PUF and on-dose authentication to guarantee a high security level for anti-counterfeiting of medicines. Specially, we report all protein-based PUFs that generate cryptographic keys with interactive multiple challenge-response pairs. The edible PUFs are made from silk (i.e., *Bombyx mori*) protein microparticles that are genetically fused with different fluorescent proteins, including enhanced cyan fluorescent protein (eCFP), enhanced green fluorescent protein (eGFP), enhanced yellow fluorescent protein (eYFP), and mKate2 (far-red) fluorescent protein. Particulate fluorescent silk is embedded in a thin film of natural white silk proteins, which can be directly attached onto the surface of a medicine in a solid oral dosage form. Resultant digitized keys are finally generated from output fluorescent images of the edible PUFs, after applying a von Neumann extractor for debiasing. To guarantees unique and unpredictable cryptographic keys with a relatively large encoding capacity, we validate the randomness of digital keys generated from the edible PUFs, using the National Institute of Standards and Technology (NIST) statistical test suite for randomness. We further characterize the fundamental PUF performance, by calculating inter-device Hamming Distances (HDs), intra-device HDs, false positive rates, and false negative rates.

## Results

**Protein-based PUFs**. Figure 1 shows that the photoluminescent properties of fluorescent silk proteins are used to realize multiple challenge-response pairs in an edible PUF platform with heightened security for on-dose authentication and anti-counterfeiting of medicines. Importantly, challenge-response pairs differentiate our protein-based PUFs from other common unique objects and tags[29,63]. In reaction to optical challenges, defined by a unique set of excitation and emission bands of different fluorescent proteins, the edible PUF made of silk protein (i.e., fibroin) and fluorescent proteins generates distinct output responses, which are used to extract digitized keys (Fig. 1a). The source of entropy is randomly distributed fluorescent silk microparticles seamlessly embedded in a covert thin transparent silk film. First, we take advantage of four different fluorescent proteins (i.e., eCFP, eGFP, eYFP, and mKate2) that have specific excitation and emission peaks in the visible wavelength range (Supplementary Table 1). Specifically, we utilize fluorescent protein-expressed silk produced by transgenic silkworms as recombinant proteins via the *piggyBac* transposase method (Supplementary Methods and Supplementary Fig. 1a). Silk proteins are an excellent biopolymer to be genetically hybridized with fluorescent protein genes[46,55,57,59]. Second, to fabricate fluorescent silk microparticles (Supplementary Fig. 2), fluorescent silk fibroin is regenerated into an aqueous

solution with a low-temperature process, is freeze-dried, and is gently ground into zeolite-shaped microparticles with sizes of $99.3 \pm 7.9 \, \mu m$ (mean ± standard deviation) (Fig. 2a, b and Supplementary Fig. 3). Third, an admixture of the fluorescent silk microparticles is broadcast on a large flat surface and a white silk fibroin solution is poured on top. After an ambient drying process in the dark, this thin transparent silk film with a thickness of 150 μm is punched into $7 \times 7 \, mm^2$ squares, resulting in all protein-based edible PUF devices (Methods and Supplementary Figs. 2 and 4). eCFP, eGFP, eYFP, and mKate2 silk cocoons possess bluish, greenish, yellowish, and reddish colors under white light illumination (Supplementary Fig. 1b). However, after the regeneration of the fluorescent silk, each type of fluorescent silk microparticles are not distinguishable in the naked eye, while maintaining their fluorescent properties (Fig. 2 and Supplementary Fig. 5). This fabrication process is scalable for mass production without using any sophisticated equipment and is safe for oral consumption without any organic solvents or synthetic polymers (e.g., methanol, ethanol, isopropanol, or polyvinyl alcohol) (Methods and Supplementary Fig. 6a). Analyses of mass spectroscopy, energy-dispersive X-ray spectroscopy, and in vitro cytotoxicity (cell viability test) assays support the overall non-toxicity of the edible PUF devices (Supplementary Methods and Supplementary Figs. 7–9).

**Cryptographic key generation**. In Fig. 3, the flow diagram illustrates how a cryptographic key is extracted from an output response when challenged by a set of excitation and emission bands, including the raw output measurement, the bitstream extraction, and the final digitized security key. We mainly use four representative challenge-response pairs ($n = 4$) based on the excitation and emission peak wavelengths of the individual fluorescent proteins in silk (Supplementary Table 1 and Supplementary Fig. 1c). An input challenge ($C_n$) is selected as a combination of the excitation and emission bands at specific wavelengths such as $\lambda_{ex} = 415 \, nm$ and $\lambda_{em} = 460 \, nm$; $\lambda_{ex} = 470$ and $\lambda_{em} = 510 \, nm$; $\lambda_{ex} = 470$ and $\lambda_{em} = 560 \, nm$; $\lambda_{ex} = 530$ and $\lambda_{em} = 630 \, nm$, corresponding to eCFP, eGFP, eYFP, and mKate2 in silk, respectively. Upon optical excitation, a raw fluorescent image is recorded by a charge-coupled device (CCD) camera equipped with a conventional zoom lens via a tunable color filter (Methods and Supplementary Fig. 10a). In other words, the corresponding fluorescent image acts as an output response ($R_n$). A resultant digitized key ($K_n$) is obtained by an extractor that converts the fluorescent image of silk microparticles to a binary bitmap (Fig. 4). First, to improve the quality of binarization, we normalize the raw fluorescent image (300 pixels × 300 pixels) by the maximum intensity (Fig. 4a). The noise is removed by applying a threshold of 20%. Fluorescent areas smaller than a specific pixel size of 20 are also considered as noise. Then, the image is resized to be 150 pixels × 150 pixels with a binning process. Second, to ensure a low bit error rate (high reproducibility), we find the spatial peak position of each fluorescent silk microparticle where the highest intensity peaks of the microparticles are located, subsequently reducing the image size to 50 pixels × 50 pixels (Fig. 4b). Then, the peak positions are only assigned to 1 bits and other pixels are 0 bits. Third, to remove the bias of 0-bits, we apply an enhanced version of the von Neumann bias compression algorithm with two-pass tuple-output debiasing (Fig. 4c)[64]. Because the fluorescent peaks are relatively rare events in the entire image due to the density of the fluorescent microparticles, global bias is present such that 0-bits are generated consistently more often than 1-bits. Finally, after debiasing, we use first 64 bits as a digitized key in each response, because a typical minimum number of peaks in the fluorescent images is 32.

**a**

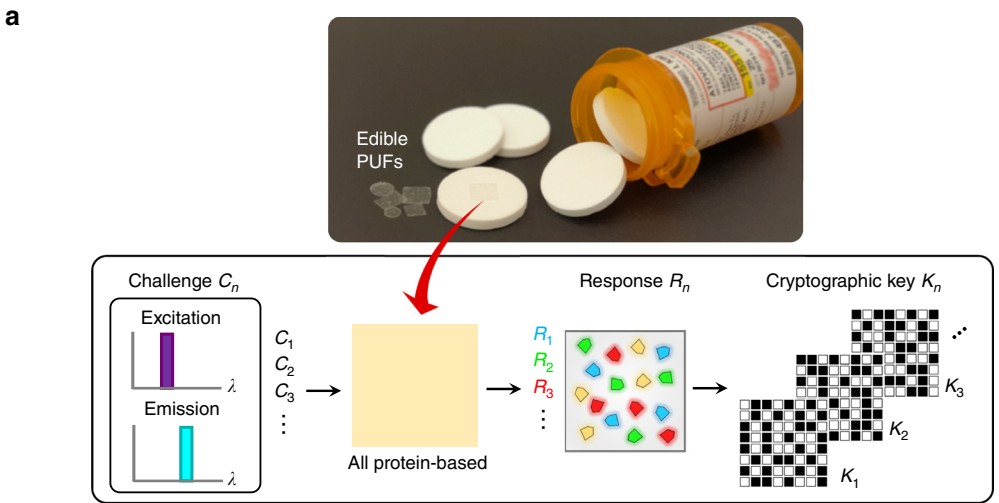

**b**

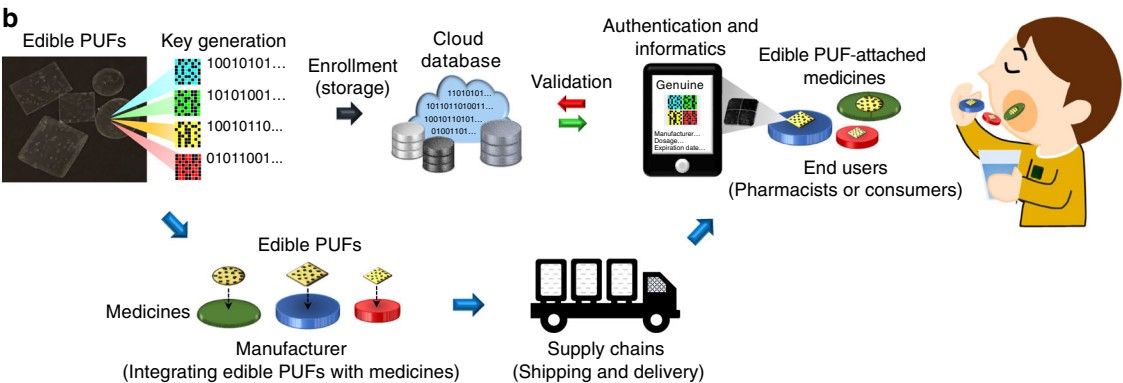

**Fig. 1 Combination of PUF and on-dose authentication for anti-counterfeiting of medicines. a** Schematic illustration of an on-dose PUF with a photograph of covert and transparent PUFs attached on the surface of medicines. The PUF device is composed of nothing but proteins from fluorescent proteins and silk to be edible and digestible. The distinct photoluminescent properties of fluorescent proteins in silk provide the parametric support of unique challenge-response pairs. In reaction to an input challenge ($C_n$), the edible PUF generates its corresponding output response ($R_n$), resulting in a cryptographic key ($K_n$). The protein-based PUFs attached on the surface of medicines can be used for on-dose authentication of each individual medicine. **b** Concept of on-dose authentication. Each individual medicine in a solid oral dosage form (e.g., tablets and capsules) is integrated with an edible PUF device by the pharmaceutical manufacturer. End users (e.g., pharmacists and consumers) can ensure the provenance and validate the medicine by accessing the enrolled digital keys in a secure database (e.g., cloud server). In addition, this edible PUF could be utilized to provide dose information and manufacturer-determined data, including product information (e.g., dosage strength, dose frequency, and expiration date), manufacturing details (e.g., location, date, batch, and lot number), and distribution path (e.g., country, distributor, wholesaler, and chain).

Combining four challenge-response pairs ($n = 4$) together, the final digitized key size results in 256 bits ($= 4 \times 64$).

We assess the quality of randomness of the edible PUF-generated binary sequences, using the NIST statistical test suite that was originally designed to evaluate random and pseudorandom number generators[65]. One of the minimal requirements of PUFs is randomness with high entropy, as PUFs rely on an entropy source to create an unclonable output response[66,67]. When PUF responses are used for cryptographic key generation, it is also critical to evaluate the randomness to ensure the unpredictability of the keys generated by PUFs. The NIST statistical test suite includes 15 different tests to quantify the randomness of bitstreams. Each test focuses on a specific aspect of randomness (Supplementary Table 2). Some of the tests rely on the minimum sequence length of $1 \times 10^6$ and the minimum number of substrings (blocks) of 55, requiring a total stream of $5.5 \times 10^7$ bits. On the other hand, the key size of the edible PUFs is significantly shorter than those of random number generators. To use seven statistical tests that require a reasonable stream

length, we explore the randomness of binary sequences summed from 30 different PUFs (Fig. 5). Specifically, we collect a total of 7680 bits from 30 different PUFs (256 bits for each PUF) and divide the bitstream into 60 sequences to perform each statistical test 60 times on individual 128-bit long sequences (Supplementary Dataset 1). Each statistical test returns two results; a p-value of a chi-squared ($\chi^2$) test and a pass rate (i.e., proportion), as shown in Table 1. As summarized in Table 1, the binary sequence from the 30 different PUFs passes all of seven NIST randomness tests without any post-processing. The parameter values used in each test and the characteristics of the NIST randomness tests are summarized in Supplementary Table 2. In other words, the bitstream (7680 bits) extracted from the 30 PUFs is statistically random, supporting the idea that the output responses of all protein-based PUFs can be unpredictable and unclonable. This result also supports the idea that our simple broadcasting process of particulate fluorescent silk offers a random spatial distribution as a straightforward yet effective entropy source[60,61].

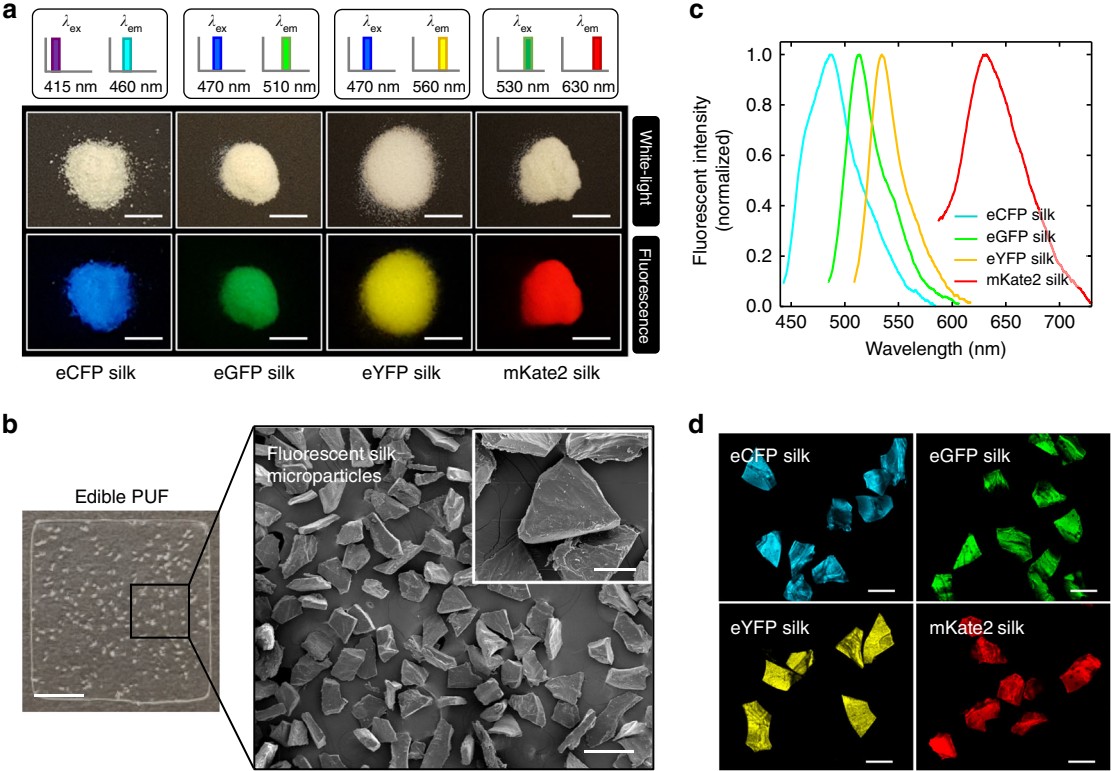

**Fig. 2 All protein-based edible PUFs made of silk and fluorescent proteins. a** Regenerated particulate eCFP, eGFP, eYFP and mKate2 silk produced by silkworm transgenesis via the *piggyBac* transposase method in which fluorescent proteins and silk (i.e., *Bombyx mori*) are genetically fused as recombinant proteins. The fluorescent images are acquired with a set of the excitation ($\lambda_{ex}$) and emission ($\lambda_{em}$) wavelengths, as specified on top of each photograph. The scale bar is 5 mm. **b** Photograph of an edible PUF device in which fluorescent silk microparticles are embedded in a thin silk film. The scale bar is 2 mm. Scanning electron microscopy (SEM) image of fluorescent silk microparticles with zeolite-like shapes. The scale bar is 200 μm. The inset shows a higher magnification SEM image of fluorescent silk microparticles. The scale bar is 50 μm. **c** The emission spectra of particulate eCFP (cyan solid line), eGFP (green solid line), eYFP (yellow solid line), and mKate2 (red solid line) silk cover a relatively broad wavelength range in the visible light, while the emission peak positions are not overlapped among others. **d** Confocal fluorescence microscopy images of the corresponding fluorescent silk microparticles under excitation wavelengths of 405, 458, 514, and 561 nm, respectively. The scale bar is 100 μm. The size of fluorescent silk microparticles is 99.3 ± 7.9 μm (mean ± standard deviation) (Supplementary Fig. 3).

**Performance characterizations of edible PUFs.** To evaluate the basic PUF performance, we examine the digitized keys of the edible PUFs. We first estimate the bit uniformity by checking the equal probability of observing 1-bit or 0-bit states:

$$\text{Bit uniformity} = \frac{1}{s} \sum_{l=1}^{s} K_l \qquad (1)$$

where $K_l$ is the $l^{th}$ binary bit of the key and $s$ is the key size. Basically, the bit uniformity is the Hamming Weight (i.e., number of 1 bits in a binary sequence) of the $s$-bit key. For 30 different PUFs, the distribution of bit uniformity converges to the ideal value of 0.5 (Fig. 6a). Then, to evaluate the device uniqueness of each PUF, we calculate an inter-device Hamming Distance (HD) by counting a number of different bits between two PUFs under the same challenge. The device uniqueness measures the degree of correlation between digitized keys measured from two different PUFs. Ideally, the digitized keys from any two selected PUF devices should be uncorrelated, indicating that the state of a PUF is unknown even when the states of other PUFs are known. The inter-device HD between any two PUF devices can be defined:

$$\text{Device uniqueness} = \frac{2}{q(q-1)} \sum_{i=1}^{q-1} \sum_{j=i+1}^{q} \frac{HD(K_i, K_j)}{s} \qquad (2)$$

where $K_i$ and $K_j$ are $s$-bit keys of the $i^{th}$ PUF device and the $j^{th}$ PUF device among $q$ different PUFs, respectively. The 30 different PUF devices generate a total of $_{30}C_2$ ($=30 \times 29/2 = 435$) comparisons. In Fig. 6b, the histogram of the normalized inter-device HDs is well fitted into a Gaussian distribution with a center at 0.5032 with a standard deviation (SD) of 0.0458, which is close to its ideal value of 0.5, exhibiting the excellent device uniqueness of all of the edible PUFs. In addition, we investigate the degree of correlation among the digitized keys of four responses in each PUF by calculating an average HD (Fig. 6c). The 30 different PUFs result in a mean HD value of 0.499 with a SD of 0.0041, indicating that the individual digitized keys in each PUF are also unique. When the number of challenge-response pairs is extended to seven, seven resultant digitized keys are still uncorrelated with a mean HD value of 0.5089 with a SD of 0.0766 (Supplementary Figs. 11 and 12).

In addition, we calculate an encoding capacity of the edible PUF-generated binary sequences. The encoding capacity simply means a number of codes that can be generated and is defined as $c^s$ where $c$ is the bit-level ($c = 2$ for binary bits of 0 and 1) and $s$ is the key size[38,68]. To accurately estimate the encoding capacity, it is important to use an appropriate key size. When an imaging scheme is used, one may think that the total number of pixels (variables) is the digitized key size. In this case, the actual encoding capacity can be less than this nominal encoding

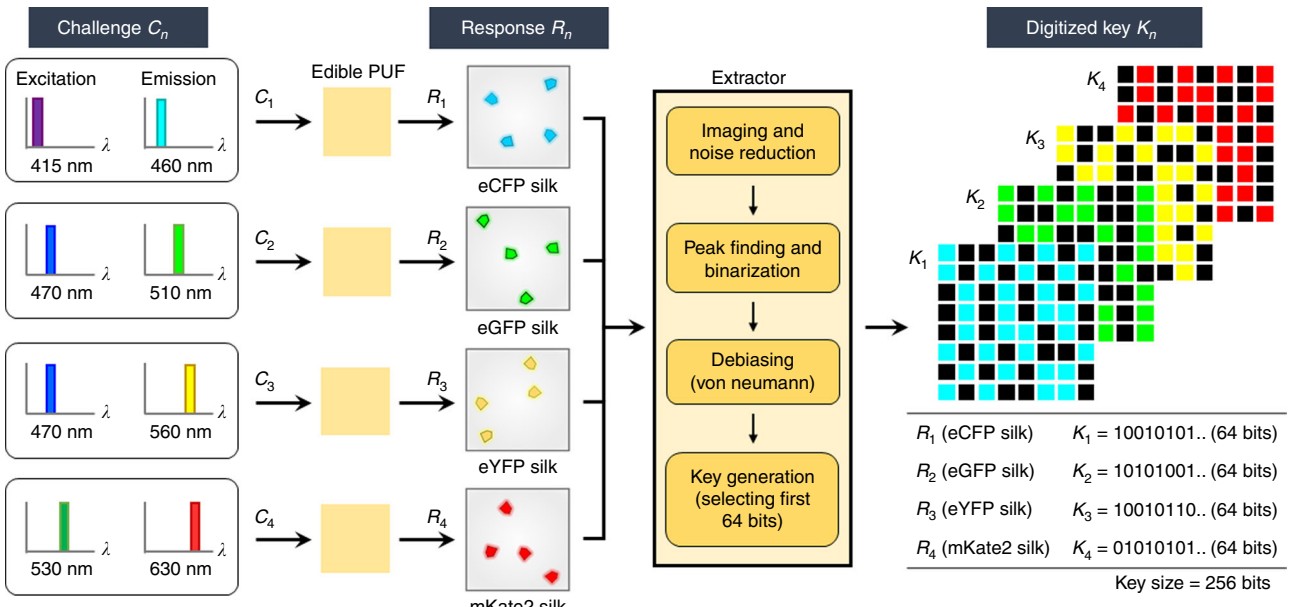

**Fig. 3 Flowchart of cryptographic key generation from an edible PUF device.** Randomly distributed fluorescent silk microparticles provide an entropy source. An input challenge ($C_n$) is determined by a set of the excitation and emission bands, which is optimized for each type of fluorescent silk (i.e., eCFP, eGFP, eYFP, and mKate2 silk). The corresponding fluorescent image acts as an output response ($R_n$). A resultant digitized key ($K_n$) is generated by undergoing four steps in the extractor. The extractor is designed to generate uniform and reproducible digital keys after data acquisitions of fluorescent images. For each response returns in a 64-bit binary key, a total key size of this PUF is 256 bits. The key size can further be enhanced by an increase in the density of fluorescent silk particles in the PUF device or by additional combinations of excitation and emission bands (Supplementary Fig. 11).

capacity, because each individual pixel (variable) cannot be completely independent. One way for estimating the number of independent pixels (variables) is to analyze the degrees of freedom; $s = p(1 - p)/\sigma^2$, where $p$ is the mean probability and $\sigma$ is the standard deviation[31,34]. In Fig. 6b, the resultant width of the inter-device HD distribution shows that significant subsets of the key are mutually independent, corresponding to the degree of freedom (or number of independent variables) of 120 ($\approx 0.5032 \times (1 - 0.5032)/0.0458^2$). As a result, the edible PUF has $c = 2$ and $s = 120$, resulting in a relatively large encoding capacity of $2^{120}$ ($\approx 1.3292 \times 10^{36}$). Importantly, the debiasing process is useful not to comprise the actual coding capacity. If a security key is biased with too many 0 s or 1 s, the actual coding capability is often diminished. A large encoding capacity could be utilized to provide information on manufacturer-determined data, including dose information (e.g., dosage strength, dose frequency, and expiration date), manufacturing details (e.g., location, date, batch, and lot number), and distribution path (e.g., country, distributor, wholesaler, and chain). If a higher encoding capacity is required for a specific application, the key size of our edible PUF can simply be scaled by further optimizing the density of fluorescent silk microparticles, which allows for a larger number of peaks in each image. The number of challenge-response pairs can also be increased by incorporating additional combinations of the excitation and emission bands (Supplementary Fig. 11).

To examine the feasibility for reliable PUFs, we test the readout reproducibility and stability of the security keys from the identical PUF device. The reproducibility of a PUF represents the ability of generating the identical keys following the same repeated challenges. We calculate an intra-device HD, which is quantitatively described by a bit error rate (i.e., percentage of error bits out of response bits with an ideal value of 0) from 10 challenge-response cycles (nine pairwise comparisons) for each PUF device. For the $i^{th}$ PUF device, an average intra-device HD captures the readout reproducibility:

$$\text{Readout reproducibility} = \frac{1}{m}\sum_{t=1}^{m}\frac{\text{HD}(K_i,\ K_{i,t})}{s} \qquad (3)$$

where $K_i$ and $K_{i,t}$ are the original $s$-bit reference key and a $s$-bit key extracted from the same PUF device at a different time-point $t$ and $m$ is the number of repeated measurements. Figure 6b shows a relatively low mean value of 0.0632 with a SD of 0.0164 estimated from the intra-device HD histogram for the 30 different PUFs at the same 10 challenge cycles. We further examine the long-term reliability under the same challenges after 60 days in the laboratory environment (i.e., stored at $22 \pm 2\,°C$ and 40–50% relative humidity in the dark) (Supplementary Fig. 13). When fluorescent intensity of the raw fluorescent images taken 60 days apart is compared, the correlation coefficients ($r$) of the four responses ($R_1$, $R_2$, $R_3$, and $R_4$) range from 0.833 to 0.983. For the pixel positions of the first 32 peaks in the binarized images, the $r$ values are even higher than 0.895 for all of the responses. These results support the potential reliability of the protein-based edible PUFs, although the reproducibility assessments do not reflect extremely harsh conditions, given the medical applications. We further estimate a false positive rate and a false negative rate from the inter-device and intra-device variabilities. When PUFs are used for authentication, the false positive rate is the probability that PUF A is authenticated as PUF B. The false negative rate is the probability that a correct PUF fails to be authenticated. The resulting false positive and false negative rates are $9.6394 \times 10^{-13}$ and $3.0982 \times 10^{-12}$, respectively, assuming that the inter-device and intra-device variabilities follow Gaussian distributions (Supplementary Fig. 14). The pairwise comparison map of cross-HD analyses further shows that all of the 30 different PUFs are highly uncorrelated (Fig. 6d) where the diagonal line indicates the intra-HD values for the identical PUF device itself,

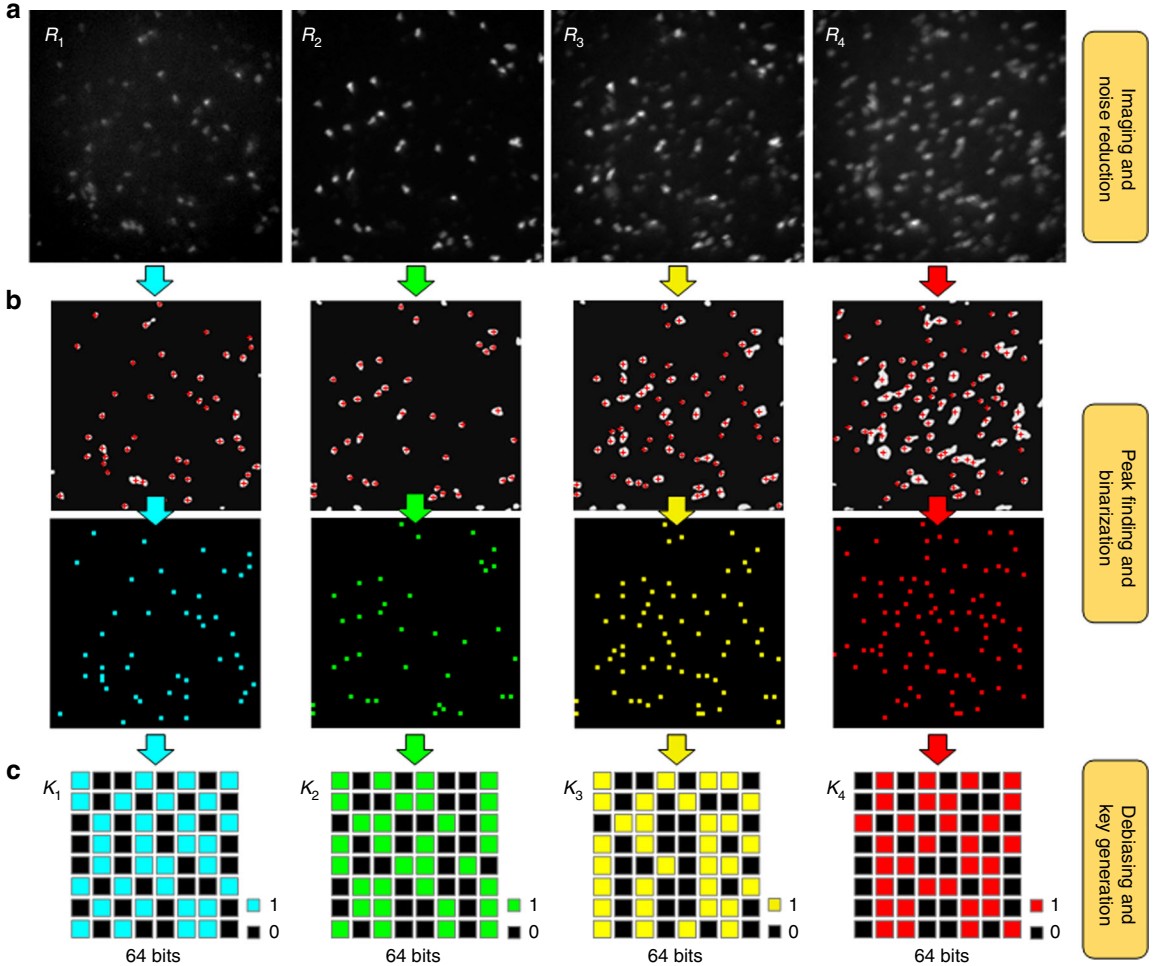

**Fig. 4 Basic elements in the extractor. a** The reading apparatus acquires raw fluorescent images of responses ($R_n$) of an edible PUF device in which an admixture of eCFP, eGFP, eYFP, and mKate2 silk microparticles is embedded in a thin silk film (300 pixels × 300 pixels). **b** The peak finding and binarization processes provide high stability and reproducibility in key generation, reducing the number of pixels to 50 pixels × 50 pixels. **c** The von Neumann debiasing process allows to compress dominant 0-bits resulting from the relatively small number of peaks. In simple von Neumann debiasing, the rate of compression is too high such that the raw data size needs to be much larger than an extracted size. In our extractor, the two-pass tuple-output von Neumann debiasing algorithm maintains a practical data size. After von Neumann debiasing, first 64 bits in each digitized key ($K_n$) are selected to create a total of 256-bit security key.

while the off-diagonal points represent the inter-HD values compared with the other PUF devices.

## Discussion

The prominent application of the reported edible PUFs is on-dose authentication to prevent patients from taking counterfeit pharmaceutical products (Fig. 1b and Supplementary Fig. 15). The edible PUF, which is flexible (Supplementary Fig. 6b, c), can be attached to flat or curved surfaces of medicines in a solid oral dosage form including pills, tablets, and capsules. Each medicine possesses unique challenge-response pairs and the end user can verify genuine or fake using the built-in flashlight LED and camera of a smartphone (Supplementary Methods and Supplementary Figs. 10b and 16) or a customized reader and accessing the registered digital keys in a database (e.g., cloud server) where each validation is guided with a trusted authority against the digital identity. In addition, the edible PUF has a self-vanishing feature. Silk proteins (i.e., fibroin) are easily be dissolved in an aqueous solution without any special treatments, owing to the disintegration property and proteolytic activity (i.e., enzymatic degradation)[47–49]. When the reported edible PUF is loaded with a

blue dye (i.e., methylene blue) for easy visual detection purpose, it is completely dissolved in deionized water after 240 min (Supplementary Fig. 17), also supporting the use for oral consumption. In other words, the end user (i.e., patient) can take the medicine without removing the PUF from the surface.

The main advantage of the reported edible PUF is an enhanced security level for on-dose authentication and anti-counterfeiting of medicines, compared to simple taggants or unique objects. First, the bitstreams generated from the edible PUF are random as verified by the NIST randomness tests. In other words, the digitized keys from the edible PUF support the unclonability originating from the spatial randomness of fluorescent microparticles via the broadcasting process. Even with the same fabrication process, it is virtually impossible to clone. Second, the multiple interactive pairs of challenges and responses in the edible PUF enhance the parametric space. In particular, the different sets of excitation and emission bands in the edible PUF provide a unique means to increase the parametric space. Importantly, each set of excitation and emission generates a unique digitized key. This enhancement in the parameter space (e.g., wavelength or frequency) would be more resistant to attempt to recreate or clone the PUF. Third, the one-time authentication feature of the

PUF device number

1      15      30

Digitized key

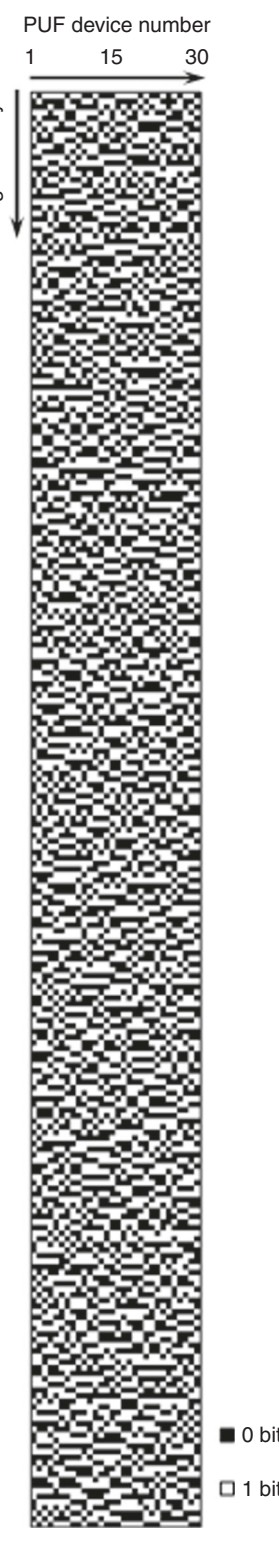

■ 0 bit

□ 1 bit

**Fig. 5 Representative binary bitmap of cryptographic keys.** Digitized security keys (256 bits in each PUF) are generated from 30 different edible PUF devices. Notably, the bitstream from this binary bitmap is random, validated by the NIST randomness tests (Table 1 and Supplementary Table 2). The production and fabrication of edible PUF devices are easily scalable due to the choice of materials and the fabrication strategy.

edible PUF can potentially rule out any unauthorized use of duplication. After a single on-dose authentication by the end user, the cryptographic key can be permanently deleted in the secure server.

**Table 1 Summary of the randomness tests of binary sequences generated from edible PUFs.**

| NIST statistical test[a] | p-value | Proportion | Result[b] |
|---|---|---|---|
| Frequency | 0.035174 | 60/60 | Pass |
| Block frequency | 0.031497 | 60/60 | Pass |
| Cumulative sums | 0.006990, 0.020085 | 60/60, 60/60 | Pass |
| Runs | 0.014216 | 59/60 | Pass |
| Longest run of ones | 0.324180 | 60/60 | Pass |
| Approximate entropy | 0.275709 | 60/60 | Pass |
| Serial | 0.232760, 0.468595 | 59/60, 60/60 | Pass |

[a]NIST tests are performed using 60 sequences of 128 bits each such that 7680 bits (i.e., digitized keys) collected from 30 different PUFs are tested. The chi-squared ($\chi^2$) distribution is used to compare the goodness-of-fit of the p-value distribution of the blocks from the entire bitstream to the expected distribution. The bitstream is considered to be random only if the p-value $\geq 0.0001$
[b]If the pass rate exceeds the minimum rate (>56/60) for each test, it is considered as a pass (Supplementary Dataset 1)

In conclusion, we have demonstrated that the integration of the natural photoluminescent proteins with the versatile optical PUF platform for on-dose authentication and anti-counterfeiting of medicines, in which multiple and interactive challenge-response pairs guarantee extremely high security. While most PUFs are primarily focused on permanent objects and devices, the reported edible PUF technology can play an important role in identification, authentication, and anti-counterfeiting where security and cryptographic protocols require immediate destruction and vanishing features (i.e., one-time authentication).

## Methods

**Materials and chemicals**. We used fluorescent silk produced by the genetic fusion of fluorescent proteins (i.e., eCFP, eGFP, eYFP, and mKate2) via germline transformation (i.e., *piggyBac* transposon) and natural white silk (Supplementary Methods)[45,46,55,57–59]. We used the following chemicals as received; dialysis tube (pore size 12,000 Da MWCO), lithium bromide (LiBr, $\geq$ 99%), miracloth (pore size 22–25 µm), sodium carbonate (Na$_2$CO$_3$, $\geq$ 99%), and Triton X100, purchased from Sigma-Aldrich Co. (Milwaukee, WI, USA). To select appropriate particle sizes and to broadcast fluorescent silk microparticles, we used two standard test sieves with opening sizes of 90 (No. 170) and 106 (No. 140) µm, purchased from Cole-Parmer (Niles, IL, USA). All experiments were performed under the ambient conditions ($22 \pm 2$ °C and $40 \pm 10$% relative humidity). It should be noted that any organic solvents and synthetic polymers were completely avoided for edibility and safe consumption. To ensure the nontoxicity of the edible PUF devices, we further conducted analyses of mass spectroscopy, energy-dispersive X-ray spectroscopy, and in vitro MTT assays (Supplementary Methods and Supplementary Figs. 7–9).

**Regeneration of transgenic fluorescent silk and white silk**. To avoid heat-induced denaturation of fluorescent proteins in silk, we performed a dissolution process of fluorescent silk under a low temperature (Supplementary Fig. 2)[69,70]. After sericin was removed, fluorescent silk cocoons were cut into small pieces with sizes less than 2–5 mm and then were dissolved in an aqueous solution of LiBr (9.5 M) at 45 °C for four hours with stirring of 400 rpm. The dissolved solution was filtered through a miracloth and was dialyzed in deionized water at room temperature for more than two days with a cellulose semipermeable tube to thoroughly remove the salt (i.e., LiBr), changing deionized water every four hours (Supplementary Methods and Supplementary Fig. 8). For eCFP, eGFP, eYFP, and mKate2 fluorescent silk, each regenerated silk fibroin solution with a concentration of 5% (w v$^{-1}$) was obtained after centrifuging with a speed of 9000 rpm at 4 °C for 20 min. The regenerated solutions were freeze-dried at $-18$ °C for seven days. The freeze-dried fluorescent silk was mechanically ground into granular microparticles with a mortar and pestle. Fluorescent silk microparticles with a size range of 90–106 µm were selected by shaking them through a stack of two standard test sieves with opening sizes of 90 and 106 µm. Similarly, a white silk fibroin solution was prepared in a dissolving solution of LiBr (9.5 M) at 60 °C for 4 h with stirring of 400 rpm, resulting in a concentration of 4% (w v$^{-1}$). It was filtered through a miracloth and was dialyzed in deionized water at room temperature for two days with a cellulose semipermeable tube, followed by centrifugation with a speed of 9000 rpm at 4 °C for 20 min.

**Fabrication of all protein-based edible PUFs**. To fabricate a large number of edible PUF devices, fluorescent silk microparticles were mixed at a ratio of 1:1:1:1

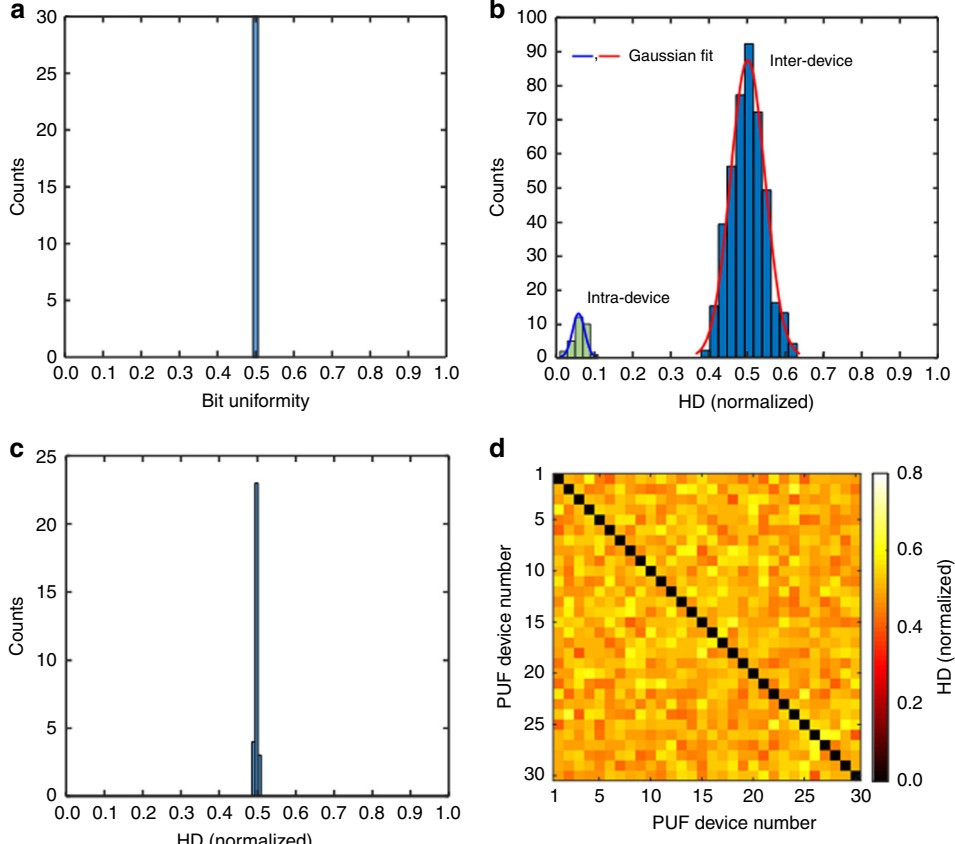

**Fig. 6 Characterizations of the basic performance matrices of edible PUFs. a** Bit uniformity calculated from 30 different PUFs shows the unbiased distribution of 0 and 1 states after von Neumann debiasing with a mean ($\mu$) value of 0.5. **b** Inter-device Hamming Distance (HD) is used to characterize the device uniqueness of 30 different edible PUFs. A Gaussian fit of the histogram returns $\mu = 0.5032$ and a standard deviation (SD; $\sigma$) of 0.0458 where the probability density of Gaussian distribution $P(x) = \frac{1}{\sigma\sqrt{2\pi}} e^{-(x-\mu)^2/(2\sigma^2)}$. The mean value is close to the ideal value of 0.5 with a narrow distribution. The mutually independent bits (degree of freedom or number of independent variables) of 120 ($\approx 0.5032 \times (1 - 0.5032)/0.0458^2$) result in an encoding capacity of $2^{120}$ ($\approx 1.3292 \times 10^{36}$). Intra-device HD is used to characterize the readout reproducibility (i.e., bit error rate) from 10 repeated challenge-response cycles (nine pairwise comparisons) in each PUF device among 30 different edible PUFs. A Gaussian fit of the histogram returns $\mu = 0.0632$ and $\sigma = 0.0164$. The clear separation between the intra-device and inter-device distributions indicates extremely low false positive and negative rates (Supplementary Fig. 14). **c** Comparisons among digitized keys from four responses in each PUF ensure the uniqueness of the responses in the PUF device. A mean HD is calculated in each PUF and 30 mean HDs from 30 different PUFs are plotted with $\mu = 0.4990$ and $\sigma = 0.0041$. **d** Pairwise comparisons among 30 different edible PUFs. The HDs in the off-diagonal areas fluctuate near the mean value of 0.5032. The source data are provided as a Source Data file.

(eCFP, eGFP, eYFP, and mKate2 silk) in a microcentrifuge tube and then was shaken by a hand. The admixture of the microparticles was broadcast to the surface of a polystyrene Petri dish (diameter 35 mm) through the sieve with an opening size of 106 μm by mechanical shaking (Supplementary Fig. 2). Subsequently, the natural white silk fibroin solution of 4 mL was poured on the plastic Petri dish and was cast at ambient conditions (25 ± 2 °C and 40–50% relative humidity) in the dark for three days, resulting in a silk film with a thickness of about 150 μm, consisting of four different fluorescent silk microparticles. Edible PUFs were prepared by punching the silk films with a square area of $7 \times 7$ mm² (Supplementary Fig. 4). It should be noted that if manufacturing of large quantities of standardized edible PUF devices is available, the final cost can further be reduced (Supplementary Methods).

**Data acquisition and reading.** As easily accessible common light sources for optical excitation, we used ultraviolet, blue, and green light-emitting diodes (LEDs) with emission wavelengths of 415 nm (FWHM = 14 nm), 470 nm (FWHM = 25 nm), and 530 nm (FWHM = 33 nm) purchased from Thorlabs Inc. (Newton, NJ, USA). Bandpass filters of 410, 470, and 530 nm (FB410-10, FB470-10, and FB530-10; Thorlabs Inc.) were placed between the light source and the PUF device. The optical power was kept at 1, 3, and 10 μW mm⁻² for 415-nm, 470-nm, and 530-nm LEDs at the surface of PUFs. To image the PUFs, we used a charge-coupled device (CCD) camera (Princeton Instruments PIXIS 1024B, Trenton, NJ, USA) with a conventional zoom lens (MVL7000, Navitar, Rochester, NY, USA) via a liquid crystal tunable filter with a FWHM of 7 nm (VariSpec VIS-07-20; PerkinElmer, Inc., Waltham, MA, USA). As a result, the following set of excitation and emission bands was selected

such that $\lambda_{ex} = 415$ nm and $\lambda_{em} = 460$ nm; $\lambda_{ex} = 470$ and $\lambda_{em} = 510$ nm; $\lambda_{ex} = 470$ and $\lambda_{em} = 560$ nm; $\lambda_{ex} = 530$ and $\lambda_{em} = 630$ nm, optimized for eCFP, eGFP, eYFP, and mKate silk, respectively.

**Image processing to extract binary keys.** To extract binary sequences from the fluorescent images of the edible PUFs, we established customized MATLAB codes for image processing and digital key extraction. Raw fluorescent images with 300 pixels × 300 pixels were normalized by the maximum fluorescent intensity in each PUF. To remove noise, an intensity threshold with 20% was applied. To improve the readout reproducibility, fluorescent areas smaller than a specific pixel size of 20 were also eliminated by the 'bwareaopen' function, and then the image was resized to 50 pixels × 50 pixels with a pixel binning process. To ensure a low bit error rate, we adopted the 'fast 2D peak finder' function to find the highest intensity peak of each fluorescent microparticle. The peaks in the images were converted into 1-bits by using the 'imbinarize' function and other pixels were assigned with 0-bits. The global bias was present such that 0-bits were generated consistently more than 1-bits, due to the low density of the fluorescent silk microparticles. To address this issue in security keys, we took advantage of the von Neumann bias compression algorithm with two-pass tuple-output debiasing[64]:

1. If the output is 00 or 11, the output is discarded.
2. If the output is 01 or 10, the first bit of 0 of 01 or 1 of 10 is retained only.
3. Reconsidering the discarded bits in the second pass, the bits are grouped as a quadruplet and the first half of the quadruplet is compared to the second half.

4. If the first half and the second half are different (i.e., 0011 or 1100), the discarded bits are retained.

It should be noted that simple von Neumann debiasing has a high rate of compression, resulting in a small key size. On the other hand, the two-pass tuple-output von Neumann debiasing algorithm maintains a practical data size. We finally selected first 64 bits in each digitized key, because the typical minimum number of the peaks in each image were 32. As a result, each PUF generates a 256-bit digital key.

**Reporting summary**. Further information on research design is available in the Nature Research Reporting Summary linked to this article.

## Data availability
The data and codes that support the findings of this study are available from the corresponding author upon reasonable request. The source data underlying Fig. 6 and Supplementary Figs. 3, 9, and 13 are provided as a Source Data file.

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

## Acknowledgements

This work was supported by the United States Air Force Office of Scientific Research (FA2386-17-1-4072), the Cooperative Research Program for Agriculture Science and Technology Development (PJ012089) from Rural Development Administration of the Republic of Korea, and the Institute for Information & Communications Technology Promotion grant (No. 2017000709) funded by the Korean government (MSIP). We thank Matthew Therkelsen and Richard Kuhn for the MTT assays.

## Author contributions

J.W.L. and Y.L.K. developed the experimental design. J.W.L. worked on the device fabrication, the optical measurements, and the key generation. M.S.K., S.H.C. and Y.M.S. participated in the analyses. S.W.K. and S.R.K. worked on transgenic silk. J.W.L., R.J.Y. and Y.L.K. wrote the paper. Y.L.K. directed the overall research. All of the authors discussed the results and the paper.

## Competing interests

J.W.L. and Y.L.K. are the inventors of provisional patent applications related to this work that have been filed to the U.S. Patents and Trademark Office by the Purdue Research Foundation (application number 62915667 filed in October 16, 2019 and application number 62915666 filed in October 16, 2019). Y.L.K. is a founding member of CryptoMED LLC. The remaining authors declare no competing interests.
