## [Peer Review File · Nature Communications]

Reviewers' Comments:

Reviewer #1:

Remarks to the Author:

This manuscript reports a plausible solution for detection of counterfeit medicines by using silk protein containing recombinant fluorescent protein to produce unique PUF devices that can generate satisfactory cryptographic keys. The distinctive feature in this study is that the PUFs are made from photonic biomaterials which can be edible and digestible. These can be attached onto the surface of medicines, enabling on-dose authentication and anti-counterfeiting in a simple and straightforward process. The authors presented a scalable process to produce PUFs and also performed experiments to evaluate the uniformity, uniqueness, randomness and reproducibility of the produced unclonable PUFs. The impressive results with ultra-low false positive/negative rates are indicative toward feasible applications. These edible protein-based PUFs would be certainly beneficial to both pharmaceutical companies and end users. I would recommend to publish this paper in your prestigious journal.

Here are some comments regarding the material and authentication method:

The report mentioned using low-temperature regeneration process to avoid heat-induced denaturation of proteins but there is concern about safety of material due to the usage of chemicals in the process such as sodium carbonate, triton X100 and lithium bromide. Has the silk protein been tested for possible trace residual amount of those chemicals? It would be good to test the toxicity of PUFs in vitro, if not in vivo at this stage, as the tag is intended for human-consumption.

The authentication method is supposed to be using smartphone camera or reader, would the light from smartphone camera be adequate to excite the fluorescent proteins to generate cryptographic keys to validate the identity of medicines under the test?

Reviewer #2:

Remarks to the Author:

This manuscript describes the use of various sized silk microparticles with corresponding fluorescent excitation and emission wavelengths as an edible physically unclonable function (PUF). Specifically, four different sets of silk microparticles (eCFP, eGFP, and mKate2) were embedded in a thin silk film. The resulting film takes advantage of the inherently random colored (different emission wavelengths) speckle pattern that also varies spatially depending on microparticle-type and location. The resulting pattern is processed into a 256-bit key. Larger keys can be constructed for larger tag sizes, which ultimately increases the number of pixels. The "on-tag" reproducibility – that is, the same key generated from the same fluorescent images – is also demonstrated.

This development of an edible PUF provides a benefit over non-consumable taggants. In fact, it is the most important aspect of this research report, which aimed to develop taggants for pharmaceuticals. However, neither the use of fluorescent materials as taggant candidates or translating unique patterns into digital keys is a unique contribution.

Recommendation: Submit to a different journal – Scientific Reports after addressing the comments with no further review required.

Comments

1. In the: (1) Abstract, "...including randomness, uniqueness, reproducibility, and low false rates."; (2) Introduction Section, "... including uniformity, uniqueness (for security), ..."; and (3) Results and Discussion Section, "... uniform as well as unique":

The authors do not clearly define what is meant by “reproducible” or “uniform” until Page 5, first continuing paragraph. The previous discussion reads as a conflict between uniformity and uniqueness. We recommend that the authors clearly establish a definition for “within-label” or “on-label” reproducibility rather than an inter-label reproducibility.

This is a nontrivial clarification as a major requirement for anti-counterfeit technologies is that the technology has a high level of security, i.e., it cannot be duplicated.

2. In the Results and Discussion Section, We recommend replacing “As the basic PUF performance” with “To evaluate the basic PUF performance”.

3. In the Results and Discussion Section, the authors state, “An input challenge (C_n) is selected as a combination of the excitation and emission bands at specific wavelengths.”

In the Introduction Section, the authors state, “ For the challenge-response requirement of PUFs, a unique set of excitation and emission bands of different fluorescent proteins serves as input challenges. The edible PUFs produce genuinely inherent output response images of spontaneous emission (i.e. fluorescence).” And, “... an output response (R_n) is obtained by an extractor...” Other places in the manuscript, they refer to the excitation being an input in which case the emission would be the output.

These are inconsistent definitions. We recommend that the authors refer to the seminal work in optical-based PUFs; this is their Reference # 36 (Pappu et al.). The R_n is not a response in the challenge-response context. The R_n is derived by post-processing the raw fluorescent image. While there are challenge-response pairs in this case due to different excitation and emission wavelengths, the response to excitation condition A is a colored speckle pattern image. Again, we recommend that the authors visit Ref #36 as well as other reports of “physical” unclonable functions to understand and correctly convey the challenge-response. This challenge-response relationship is independent of further processing.

4. In the Results and Discussion Section, the authors state, “In other words, the bitstream (7,680 bits) extracted from the 30 PUFs is statistically random, supporting the idea that the output responses of all protein-based PUFs can be unpredictable and unclonable.” Again, a major requirement for anti-counterfeit technologies is that the technology has a high level of security, i.e., it cannot be duplicated. The rationale for unclonability provided states that the keys extracted from 30 silk microparticle “cocktails” was random. The prevention of duplication refers more to being able to duplicate the response of the “system/device”. For example, if a counterfeiter had access to a taggant, could they recreate the spatial and color pattern? These are microparticles, so it is reasonable to think that these particles could be duplicated and manipulated into duplicate positions. We recommend that the authors establish a stronger rationale for duplication prevention and/or discuss the level of difficulty for microparticle synthesis and spatial positioning would be.

5. In the Results and Discussion Section, the authors state, “... the end user can verify genuine or fake using a smartphone camera or a customized reader...”; however, there are no examples of fluorescent images taken with a smartphone. We recommend that the authors demonstrate the feasibility of capturing the various colored patterns with a smartphone and adding this image to the Supplemental Information. We also recommend that the authors mention the difficulties with having an end-user capture these sorts of images.

6. In Figure Caption of Figure 1:

Change “convert” to “covert”

Change “In reaction by” to “in reaction to”

7. In Figure Caption of Figure 6:

This is an incomplete sentence: “... As each individual...”. We also recommend that the authors make this Figure their first Figure to clarify the overall concept immediately.

8. It is unclear what the authors meant by, "... to perform each statistical 60 times..." (in Results and Discussion Section).

Reviewer #3:

Remarks to the Author:

This manuscript outlines the use of fluorescent proteins and silk proteins as a way to create edible photonic biomaterials for on-dose authentication of individual pills. This platform has the potential to address a critical need and from a materials science perspective the system is well designed. As such, I recommend the manuscript be revised for publication in accordance to the following specific comments.

1) This field is highly interdisciplinary and there is quite a bit of jargon in the analysis of the PUF section - the authors are encouraged to really make sure that terms are defined for a general audience.

2) How feasible is this platform from a cost perspective? How much material would be required per surface area to create a reliable PUF?

3) Many pills have curved surfaces. Could this be used on a curved surface and analyzed? How about if coated with a polymer encapsulant?

4) My biggest concern with the manuscript with the discussion of how the encoding capacity is calculated. It simply doesn't make any sense and doesn't conform to standard notation. This portion of the manuscript really needs to be simplified and clarified for a broader audience.

Responses to Reviewer #1

General comment

This manuscript reports a plausible solution for detection of counterfeit medicines by using silk protein containing recombinant fluorescent protein to produce unique PUF devices that can generate satisfactory cryptographic keys. ... The impressive results with ultra-low false positive/negative rates are indicative toward feasible applications. These edible protein-based PUFs would be certainly beneficial to both pharmaceutical companies and end users. I would recommend to publish this paper in your prestigious journal.

Our response

We are grateful for the enthusiastic support from Reviewer #1.

Specific comment 1

The report mentioned using low-temperature regeneration process to avoid heat-induced denaturation of proteins but there is concern about safety of material due to the usage of chemicals in the process such as sodium carbonate, triton X100 and lithium bromide. Has the silk protein been tested for possible trace residual amount of those chemicals? It would be good to test the toxicity of PUFs in vitro, if not in vivo at this stage, as the tag is intended for human-consumption.

Our response

We agree with the comment that it is critical for us to double-check the toxicity. We have conducted additional analyses to ensure the nontoxicity of our edible PUFs, using mass spectroscopy (Supplementary Fig. 7 in the revised paper), energy-dispersive X-ray (EDX) spectroscopy (Supplementary Fig. 8 in the revised paper), and MTT assays (Supplementary Fig. 9 in the revised paper). First, the result from mass spectroscopy shows that there are no residual components of Triton X100, which was used in the removal of sericin (i.e. degumming). Second, the result from EDX spectroscopy shows that there is no detectable trace of bromine from lithium bromide, which was used to dissolve silk cocoons as a special type of salt. Finally, the MTT assay result shows that the metabolic activity of the silk samples is not different from that of controls (i.e. no silk) with a p -value of 0.81, ruling out the general cytotoxicity (i.e. cell viability). It should be noted that sodium carbonate (Na_2CO_3 and also known as soda crystals), which was used in the degumming process, is an inactive ingredient for drug products approved by FDA (Supplementary Ref. 11). In addition, it is well known that natural silk also contains the elements of carbon (C), oxygen (O), and sodium (Na). As suggested, we have included these additional results in the revised paper.

Specific comment 2

The authentication method is supposed to be using smartphone camera or reader, would the light from smartphone camera be adequate to excite the fluorescent proteins to generate cryptographic keys to validate the identity of medicines under the test?

Our response

Yes, it is straightforward to use the built-in flashlight LED and camera of conventional smartphones. Accordingly, we have performed additional experiments using a smartphone (Supplementary Figs. 10b and 16 in the revised paper). Because the built-in flashlight LED in an Android smartphone (Samsung Galaxy Note 9) has a strong blue light component at the wavelength $\lambda = 450$ nm, a shortpass filter (cut-off at 450 nm) can easily convert the flashlight LED to an excitation light source for fluorescence imaging. We have further demonstrated the digitized key extraction using the built-in camera. As suggested, we have included these results in the revised paper.

Responses to Reviewer #2

General comment

This manuscript describes the use of various sized silk microparticles with corresponding fluorescent excitation and emission wavelengths as an edible physically unclonable function (PUF). ... Larger keys can be constructed for larger tag sizes, which ultimately increases the number of pixels. The “on-tag” reproducibility – that is, the same key generated from the same fluorescent images – is also demonstrated. This development of an edible PUF provides a benefit over non-consumable taggants. In fact, it is the most important aspect of this research report, which aimed to develop taggants for pharmaceuticals. ...

Our response

We are grateful for the support from Reviewer #2. To the best of our knowledge, our paper introduces two novel strategies for anti-counterfeiting of medicines: i) physical unclonable functions (PUFs) and ii) ‘on-dose’ (or in-dose) authentication. As originally introduced in hardware security, PUFs should be differentiated from simple taggant technologies, in part because PUFs provide an enhanced parametric space with multiple challenge-response pairs. ‘On-dose’ authentication allows every individual pill or dose to be verified as genuine and then the patient can simply ingest it. In other words, our reported security technology is the first-of-a-kind of combining the concept of PUF and all protein-based edible materials for ‘on-dose’ authentication.

Specific comment 1

The authors do not clearly define what is meant by “reproducible” or “uniform” until Page 5, first continuing paragraph. We recommend that the authors clearly establish a definition for “within-label” or “on-label” reproducibility rather than an inter-label reproducibility. This is a nontrivial clarification as a major requirement for anti-counterfeit technologies is that the technology has a high level of security, i.e., it cannot be duplicated.

Our response

As the reviewer mentioned, we provided the definitions of uniformity and uniqueness throughout the original paper. As suggested, we have added additional words, such as ‘bit’ uniformity, ‘device’ uniqueness, and ‘readout’ reproducibility. The key definitions can be recapitalized as follows: The ‘bit’ uniformity in each PUF device is defined as the equal probability of observing ‘1’- or ‘0’-bit states (Eq. 1 in the original paper). The ‘device’ uniqueness (inter-device) meant that the degree of correlation among different PUF devices is extremely low (Eq. 2 in the original paper). The ‘within-label’ or ‘on-label’ reproducibility (intra-device) that the reviewer mentioned was already included in the original paper such that the reproducibility of the identical PUF is defined as the ability of generating the identical digitized key (Eq. 3 in the original paper). As the reviewer indicated, researchers in different security domains have used different terms, although the basic meanings behind are the same. Fortunately, the recent advances in PUF have helped to bring a consensus. One of the examples is the recent paper (“PUF taxonomy,” *Applied Physics Reviews* 6:011303, 2019). In this respect, we used the standard PUF terms that the general security community uses.

Specific comment 2

In the Results and Discussion Section, We recommend replacing “As the basic PUF performance” with “To evaluate the basic PUF performance”.

Our response

As suggested, we have changed it.

Specific comment 3

In the Results and Discussion Section, the authors state, “An input challenge (Cn) is selected as a combination of the excitation and emission bands at specific wavelengths.” In the Introduction Section, the authors state, “For the challenge-response requirement of PUFs, a unique set of excitation and emission bands of different fluorescent proteins serves as input challenges. The edible PUFs produce genuinely inherent output response images of spontaneous emission (i.e. fluorescence).” And, “... an output response (Rn) is obtained by an extractor...” ... We recommend that the authors refer to the seminal work in optical-based PUFs; this is their Reference # 36 (Pappu et al.).

Our response

We thank the reviewer for this particular comment. As previously explained to the editor, the concept of our edible PUF stemmed from the seminal work on the optical PUF (Reference # 36 by Pappu et al: “Physical one-way functions,” *Science* 297:2026, 2002). Although we conducted the similar analyses, our realization of challenge-response pairs is completely different. In the original optical PUF, a separate input with a different incident angle of the laser beam was used as a different challenge, but the output responses or digitized keys were often correlated among others to an extent.

In our case, a unique fluorescent image is formed by a different combination of excitation and emission bands. In other words, a unique set of excitation and emission bands serves as an input challenge (C_n). The corresponding fluorescent image acts as an output response (R_n). A resultant digitalized key (K_n) is finally extracted from the output image. Importantly, the digitalized key from each response within the same PUF is highly uncorrelated (Fig. 5c in the original paper and Fig. 6c in the revised paper). As suggested, we have kept this notation in the revised paper.

Specific comment 4

... Again, a major requirement for anti-counterfeit technologies is that the technology has a high level of security, i.e., it cannot be duplicated. The rationale for unclonability provided states that the keys extracted from 30 silk microparticle “cocktails” was random. The prevention of duplication refers more to being able to duplicate the response of the “system/device”. For example, if a counterfeiter had access to a taggant, could they recreate the spatial and color pattern? These are microparticles, so it is reasonable to think that these particles could be duplicated and manipulated into duplicate positions. We recommend that the authors establish a stronger rationale for duplication prevention and/or discuss the level of difficulty for microparticle synthesis and spatial positioning would be.

Our response

We thank the reviewer for asking this general issue, because this allows us to bring up the novelty of our approach. Indeed, this is the very reason why we introduced the concept of PUFs in the pharmaceutical community. PUFs are centered in increasing the parametric space by implementing multiple challenge-response pairs. As the reviewer mentioned, in the case of a simple taggant, a counterfeiter might be able to read the locations of microparticles and clone them. In the reported edible PUFs, it is further required to figure out all of the sets of excitation and emission bands for each fluorescent microparticle. This significant enhancement in the parameter space (e.g. wavelength or frequency) would be more resistant to attempt to recreate or clone the PUF. Indeed, our main advantage of the edible PUF is an enhanced security level for ‘on-dose’ authentication and anti-counterfeiting of medicines.

The high security features of this reported edible PUF can be recapitalized as follows: First, the bitstreams generated from the edible PUF are random as verified by the NIST randomness tests. One of the minimal requirements of PUF is randomness with high entropy, as PUFs rely on an entropy source to create an unclonable output response. In other words, the digitized keys from the edible PUF support the unclonability originating from the spatial randomness of fluorescent microparticles via the broadcasting process. Even with the same fabrication process, it is virtually impossible to clone. Second, the multiple interactive pairs of challenges and responses in the edible PUF enhance the parametric space, which is advantageous over simple taggants or unique objects. In particular, the different sets of excitation and emission bands in the edible PUF provide a unique means to increase the parametric space. Importantly, each set of excitation and emission generates a unique digitized key. Third, the one-time authentication feature of the edible PUF can potentially rule out any unauthorized use of duplication. After a single ‘on-dose’ authentication, the cryptographic key can be permanently deleted in the secure server, preventing a cloned PUF from being used. As suggested, we have included these high-security attributes in the revised paper.

Specific comment 5

In the Results and Discussion Section, the authors state, “... the end user can verify genuine or fake using a smartphone camera or a customized reader...”; however, there are no examples of fluorescent images taken with a smartphone. We recommend that the authors demonstrate the feasibility of capturing the various colored patterns with a smartphone and adding this image to the Supplemental Information. We also recommend that the authors mention the difficulties with having an end-user capture these sorts of images.

Our response

As suggested, we have included a feasibility study using the built-in LED and camera of a conventional smartphone (Supplementary Figs. 10b and 16 in the revised paper).

Specific comments 6 – 8

6. In Figure Caption of Figure 1: Change “convert” to “covert” Change “In reaction by” to “in reaction to”

7. This is an incomplete sentence: “... As each individual...”. We also recommend that the authors make this Figure their first Figure to clarify the overall concept immediately.

8. It is unclear what the authors meant by, “... to perform each statistical 60 times...” (in Results and Discussion Section).

Our response

As suggested, we have revised them in the revised paper.

Responses to Reviewer #3

General comment

This manuscript outlines the use of fluorescent proteins and silk proteins as a way to create edible photonic biomaterials for on-dose authentication of individual pills. This platform has the potential to address a critical need and from a materials science perspective the system is well designed.

Our response

We are truly grateful for the enthusiastic support from Reviewer #3.

Specific comment 1

This field is highly interdisciplinary and there is quite a bit of jargon in the analysis of the PUF section - the authors are encouraged to really make sure that terms are defined for a general audience.

Our response

We completely agree with the reviewer's comment. Researchers in different security domains have used different terms, although the basic meanings behind are the same. The recent advances in PUF have helped to bring a consensus, as summarized in the recent paper ("PUF taxonomy," *Applied Physics Reviews* **6**:011303, 2019). In our original paper, we tried to stick to the standard terms that the general security community understands. As suggested, we have added the basic definitions for the general and broad readers of *Nature Communications*.

Specific comment 2

How feasible is this platform from a cost perspective? How much material would be required per surface area to create a reliable PUF?

Our response

As suggested, we have included a cost analysis in detail (Supplementary Methods). The material cost of the edible PUF can be estimated to be \$0.01 – \$0.02 per item. This corresponds to 50 – 60 mg of silk fibroin in each PUF. Overall, this cost would be negligible, compared to personalized drugs, typical lifestyle drugs, and lifesaving drugs. If manufacturing of large quantities of standardized edible PUF is available, the final cost can further be reduced.

Specific comment 3

Many pills have curved surfaces. Could this be used on a curved surface and analyzed? How about if coated with a polymer encapsulant?

Our response

We have tested that the edible PUF can easily be integrated with curved surfaces, using FDA-approved 000-size gelatin capsules (Supplementary Fig. 6c in the revised paper).

Specific comment 4

My biggest concern with the manuscript with the discussion of how the encoding capacity is calculated. It simply doesn't make any sense and doesn't conform to standard notation. This portion of the manuscript really needs to be simplified and clarified for a broader audience.

Our response

The encoding capacity simply means a number of codes that can be generated. Technically, the encoding capacity is defined as c^s where c is the bit-level ($c = 2$ for binary bits of '0' and '1') and s is the key size (*Nature Biotechnology* **19**:631, 2001; *Science Advances* **4**:e1701384, 2018). Indeed, this encoding capacity calculation is extensively used in the PUF community. To accurately estimate the encoding capacity, it is important to use an appropriate key size. When an imaging scheme is used, one may think that the total number of pixels (variables) is the digitized key size. In this case, the actual encoding capacity can be less than this nominal encoding capacity, because each individual pixel (variable) cannot be completely independent. One way for estimating the number of independent pixels (variables) is to analyze the distribution based on the central limit theorem (also known as the degree of freedom); $s = p(1 - p)/\sigma^2$, where p is the mean probability and σ is the standard deviation (e.g. *Science* **297**:2026, 2002; *Nature Nanotechnology* **11**:559, 2016). Accordingly, we used the number of independent pixels in estimating the actual encoding capacity. As suggested, we have added this basic explanation in the revised paper for the general and broad readers of *Nature Communications*.

Reviewers' Comments:

Reviewer #1:

Remarks to the Author:

Authors have carefully revised the manuscript to address our concerns with the new data provided.

Reviewer #2:

Remarks to the Author:

Overall, the changes and additions are satisfactory. We recommend publication without further changes.

Of note:

For Reviewer 2, Specific Comment 1: The authors do not clearly define what is meant by "reproducible" or "uniform" until Page 5, first continuing paragraph. We recommend that the authors clearly establish a definition for "within-label" or "on-label" reproducibility rather than an inter-label reproducibility. This is a nontrivial clarification as a major requirement for anti-counterfeit technologies is that the technology has a high level of security, i.e., it cannot be duplicated.

We appreciate the additional detail provided by the authors; however, we were merely requesting to make this clarification earlier in the manuscript rather than waiting until Page 5.

For Reviewer 2, Specific Comment 4: This was a crucial addition to the manuscript. The clarification that the duplication relies on not only the physical PUF itself (which could be duplicated regardless of the difficulty and/or expertise needed) but also on the challenge-response pairs selected. The latter is information a counterfeiter presumably would not have access to. We thank you for this addition.

Reviewer #3:

Remarks to the Author:

I am satisfied with the revisions.

Reviewer #1

Comment

Authors have carefully revised the manuscript to address our concerns with the new data provided.

Our response

We are grateful for the enthusiastic support from Reviewer #1.

Reviewer #2

Specific comment regarding the previous comment 1

[This is the previous comment 1] The authors do not clearly define what is meant by “reproducible” or “uniform” until Page 5, first continuing paragraph. We recommend that the authors clearly establish a definition for “within-label” or “on-label” reproducibility rather than an inter-label reproducibility. This is a nontrivial clarification as a major requirement for anti-counterfeit technologies is that the technology has a high level of security, i.e., it cannot be duplicated.

***We appreciate the additional detail provided by the authors;** however, we were merely requesting to make this clarification earlier in the manuscript rather than waiting until Page 5.*

Our response

As suggested, we have included the basic definitions in the Introduction in a brief manner, including an additional reference.

Specific comment regarding the previous comment 4

*This was a crucial addition to the manuscript. The clarification that the duplication relies on not only the physical PUF itself (which could be duplicated regardless of the difficulty and/or expertise needed) but also on the challenge-response pairs selected. The latter is information a counterfeiter presumably would not have access to. **We thank you for this addition.***

Our response

We are grateful for the enthusiastic support from Reviewer #2.

Reviewer #3

Comment

I am satisfied with the revisions.

Our response

We are grateful for the enthusiastic support from Reviewer #3.